# Are Changes in the Percentage of Specific Leukocyte Subpopulations Associated with Endogenous DNA Damage Levels in Testicular Cancer Patients?

**DOI:** 10.3390/ijms22158281

**Published:** 2021-07-31

**Authors:** Katarina Kalavska, Zuzana Sestakova, Andrea Mlcakova, Katarína Kozics, Paulina Gronesova, Lenka Hurbanova, Viera Miskovska, Katarina Rejlekova, Daniela Svetlovska, Zuzana Sycova-Mila, Jana Obertova, Patrik Palacka, Jozef Mardiak, Michal Chovanec, Miroslav Chovanec, Michal Mego

**Affiliations:** 1Translational Research Unit, Faculty of Medicine, Comenius University, National Cancer Institute, 833 10 Bratislava, Slovakia; katarina.hainova@gmail.com (K.K.); daniela.svetlovska@nou.sk (D.S.); 2Department of Molecular Oncology, Cancer Research Institute, Biomedical Research Center, Slovak Academy Sciences, 845 05 Bratislava, Slovakia; 3Department of Genetics, Cancer Research Institute, Biomedical Research Center, Slovak Academy Sciences, 845 05 Bratislava, Slovakia; zuzana.sestakova@savba.sk (Z.S.); lenka.hurbanova@savba.sk (L.H.); miroslav.chovanec@savba.sk (M.C.); 4Department of Hematology, National Cancer Institute, 833 10 Bratislava, Slovakia; andrea.mlcakova@nou.sk; 5Department of Nanobiology, Cancer Research Institute, Biomedical Research Center, Slovak Academy Sciences, 845 05 Bratislava, Slovakia; katarina.kozics@savba.sk; 6Department of Immunology, Cancer Research Institute, Biomedical Research Center, Slovak Academy Sciences, 845 05 Bratislava, Slovakia; paulina.gronesova@gmail.com; 71st Department of Oncology, Faculty of Medicine, Comenius University, St. Elisabeth Cancer Institute, 812 50 Bratislava, Slovakia; vieramiskovska@yahoo.fr; 82nd Department of Oncology, Faculty of Medicine, Comenius University, National Cancer Institute, 833 10 Bratislava, Slovakia; katarina.rejlekova@nou.sk (K.R.); michal.chovanec1@gmail.com (M.C.); 9Department of Oncology, National Cancer Institute, 833 10 Bratislava, Slovakia; zuzana.sycova-mila@nou.sk (Z.S.-M.); jana.obertova@nou.sk (J.O.); patrik.palacka@nou.sk (P.P.); jozef.mardiak@nou.sk (J.M.)

**Keywords:** germ cell tumors, tumor microenvironment, immune microenvironment, DNA damage level

## Abstract

Chemoresistance of germ cell tumors (GCTs) represents an intensively studied property of GCTs that is the result of a complicated multifactorial process. One of the driving factors in this process is the tumor microenvironment (TME). Intensive crosstalk between the DNA damage/DNA repair pathways and the TME has already been reported. This study aimed at evaluating the interplay between the immune TME and endogenous DNA damage levels in GCT patients. A cocultivation system consisting of peripheral blood mononuclear cells (PBMCs) from healthy donors and GCT cell lines was used in an in vitro study. The patient cohort included 74 chemotherapy-naïve GCT patients. Endogenous DNA damage levels were measured by comet assay. Immunophenotyping of leukocyte subpopulations was performed using flow cytometry. Statistical analysis included data assessing immunophenotypes, DNA damage levels and clinicopathological characteristics of enrolled patients. The DNA damage level in PBMCs cocultivated with cisplatin (CDDP)-resistant GCT cell lines was significantly higher than in PBMCs cocultivated with their sensitive counterparts. In GCT patients, endogenous DNA damage levels above the cutoff value were independently associated with increased percentages of natural killer cells, CD16-positive dendritic cells and regulatory T cells. The crosstalk between the endogenous DNA damage level and specific changes in the immune TME reflected in the blood of GCT patients was revealed. The obtained data contribute to a deeper understanding of ongoing interactions in the TME of GCTs.

## 1. Introduction

Germ cell tumors (GCTs) are the most common solid malignancy among young men aged 14–44 years. Incidence has progressively increased over the last two decades, especially in Western countries [1,2]. GCTs are traditionally considered “a model for cure” because the cure rate reaches >90%. Despite the exceptional treatment effect of the combination of surgery and cisplatin (CDDP)-based chemotherapy, there is a subgroup of patients (approximately 10–15%) who become refractory to therapy or experience disease relapse. The prognosis of this smaller subgroup is dismal, and effective treatments are still lacking [1,3].

The chemoresistance of GCT is a complex and multifactorial phenomenon that is closely associated with the tumor microenvironment (TME) [4]. Mammalian testes are characterized by a special immunological environment marked as the “immune privilege status” [5]. At present, published data indicate that GCTs are richly infiltrated by immune cells that modulate the TME in multiple ways, including cytokine secretion. The interplay between tumor-infiltrating immune cells and tumor cells creates favorable conditions for tumor survival and expansion [4,6]. Siska and colleagues provided interesting data characterizing T cell subsets and immune checkpoints in the immune infiltrate in testicular GCTs by multiplexed fluorescence immunohistochemistry (FIHC). The advanced stage of disease was related to changes in immune cell infiltration, irrespective of tumor histology. Namely, T and natural killer (NK) cell populations associated with antitumor immunity were decreased, while potentially protumor immune populations, including regulatory T cells (Tregs), neutrophils, mast cells, and macrophages, were significantly elevated [7]. In addition, it has been shown that GCTs, particularly seminomas, are richly infiltrated by immune cells [8,9,10]. The TME of GCTs is modulated by specific cytokine patterns generating proangiogenic activity in tumors. T and NK cell stimulation is reduced, which ultimately results in the immune response inhibition [11]. Moreover, Pearce et al. described strong CD4+ and CD8+ cancer/testis antigen (CTAg)-specific T cell responses in the peripheral blood (PB) of patients with GCTs that indicate the presence of CTAg-specific immunity consisting of short-lived effector T cells in these patients [12]. In addition, it has been suggested that immune processes play a role in the progression of disease because an increased systemic immune inflammation index (SII), calculated as platelet times neutrophil/lymphocyte counts from PB smears prior to chemotherapy, is significantly correlated with worse prognosis in GCT patients [13]. Furthermore, the SII and neutrophil-to-lymphocyte ratio (NLR) were reported as independent predictors of progression-free survival (PFS) and overall survival (OS) in a cohort of 62 relapsed/refractory GCT patients undergoing high-dose chemotherapy (HDCT). The SII, NLR and platelet-to-lymphocyte ratio (PLR) were also significantly associated with the overall response to HDCT [14].

Recently, emerging evidence has indicated an intensive interplay between DNA damage/DNA repair pathways and the TME [15]. Our previous work [16,17] revealed that the levels of endogenous DNA damage, measured by the comet assay in PBMCs, is an independent prognostic factor in GCTs. Chemotherapy-naïve GCT patients with % DNA in the tail above the cutoff value of 6.34 had significantly worse PFS and OS than did those patients with % DNA in the tail below or equal to the cutoff value. Furthermore, the DNA damage levels correlated with the metastatic process in GCT patients in terms of both the presence of metastases and the number of metastatic sites. The association between higher endogenous DNA damage levels and an increased risk of acute hematological toxicity with significantly lower nadirs of white blood cells, absolute neutrophil and lymphocyte counts were also documented [18].

Based on our previous data, we further examined the possibility that there is a link between the specific immune cell populations and the levels of endogenous DNA damage measured in PBMCs from chemotherapy-naïve GCT patients. First, we cocultured PBMCs isolated from healthy donors with GCT cell lines in vitro. Subsequently, a more comprehensive analysis of patient samples was carried out. Data obtained herein may contribute to elucidating the biological pathways underlying the prognostic value of endogenous DNA damage levels in peripheral lymphocytes in patients with chemotherapy-naïve testicular GCTs [16,17]. A more thorough understanding of these pathways may help to better stratify GCT patients with a high risk for relapse or poor prognosis and identify new therapeutic targets.

## 2. Results

### 2.1. 2D Cocultivation of GCT Cell Lines with PBMCs

2D coculture model system was used to investigate possible interaction between PBMCs and tumor cells and their CDDP-resistant variants, respectively, and so partially mimic the interplay between tumor cells and PBMCs in GCTs patients. The genotoxic effects resulting from coculturing GCT cell lines with PBMCs were evaluated by the comet assay and expressed as a % of tail DNA. Applying the 2D coculture model, we have achieved interesting results showing that the DNA damage level in PBMCs cocultivated with cisplatin (CDDP)-resistant GCT cell lines is significantly higher than in PBMCs cocultivated with their sensitive counterparts after 120 h of cocultivation. These data suggest that CDDP-resistant tumor cells interact with PBMCs in a different way, when compared to cisplatin-sensitive tumor cells. However, we also accept the incomplete approximation of the in vitro setups that are not fully able to simulate TME when compared to the 3D organoid model system (see the limitations of the study).

Initially, DNA damage levels were measured in PBMCs after 24, 48, 72, 96 and 120 h of coculture with GCT cell lines (data not shown). Since virtually no difference in DNA damage level was observed in PBMCs cocultured for 24, 48 and 72 h with CDDP-resistant cells compared to those cultured with CDDP-sensitive cells, only time points 96 and 120 h were used in further analyses. The time point 120 h was also established as the terminal time point, when we were able to cultivate GCT cell lines without passaging.

We determined significantly higher (*p* < 0.01) DNA damage levels in PBMCs coculti-vated with the CDDP-resistant yolk sac tumor cell line NOY-1 CisR than in PBMCs cocul-tivated with CDDP-sensitive NOY-1 cells at the 120-h time point. Similarly, the DNA damage level in PBMCs cocultivated for 120 h with the CDDP-resistant embryonal carcinoma cell line NTERA-2 CisR or the seminoma cell line TCam-2 CisR was significantly higher than in PBMCs cocultivated with their sensitive counterparts (*p* < 0.05) (Figure 1).

### 2.2. GCT Subjects’ Analyses

#### 2.2.1. Patient’s Characteristics

The patients’ clinicopathological characteristics are summarized in Table 1. The median age of the patients was 35 years (range 19–62 years). The majority of patients had a nonseminoma histology of tumors (73.0%) and a good prognosis (58.1%) according to the International Germ Cell Cancer Collaborative Group (IGCCCG) [19]. Metastatic disease was present in 61 (82.4%) of patients.

#### 2.2.2. Associations among the Clinicopathological Characteristics, Percentage of Different Leukocyte Subsets and Endogenous DNA Damage Level

Our analysis revealed that endogenous DNA damage levels above the cutoff value are independently associated with increased percentages of NK cells, CD16-positive DCs and Tregs.

Of 74 patients, 19 (25.7%) had a DNA damage level ≤6.34, and 55 (74.3%) patients displayed a DNA damage level >6.34. Using univariate analysis, we found no association between the DNA damage level and any of the patients’ clinicopathological characteristics (Table 2).

Immunophenotyping of selected leukocyte subpopulations was performed using the specific flow cytometry gating strategy (Figure 2).

However, univariate analysis revealed an inverse correlation between the percentage of B cells and the level of endogenous DNA damage. The mean percentage (±standard error of the mean; SEM) of B cells was significantly lower in patients with DNA damage levels above the cutoff value of 6.34 (8.3 ± 1.2% vs. 12.6 ± 0.7%, *p* = 0.00058). Patients with endogenous DNA damage levels higher than the cutoff had a significantly higher percentage of NK cells (17.3 ± 1.9% vs. 11.3 ± 1.1%, *p* = 0.008). Similarly, a positive correlation was determined between endogenous DNA damage and the percentage of CD16-positive DCs (64.1 ± 6.5% vs. 41.5 ± 2.8%, *p* = 0.00574). Moreover, patients with DNA damage levels ≤ 6.34 had a significantly elevated percentage of Tregs compared to patients with DNA damage levels below the cutoff value (4.5 ± 0.3% vs. 3.8 ± 0.2%, *p* = 0.03937). In addition, this association was confirmed in a subpopulation of Tregs derived from CD4+ lymphocytes (10.4 ± 0.6% and 8.4 ± 0.4%, respectively; *p* = 0.01637) (Table 3).

#### 2.2.3. Multivariate Associations among the Patients’ Clinicopathological Characteristics, Percentage of Different Leukocyte Subpopulations and Endogenous DNA Damage Level

In the multivariate analysis, the percentages of NK cells, DCs and Tregs were independently related to endogenous DNA damage levels. Multivariate analysis did not confirm the percentage of B cells as an independent factor associated with the DNA damage level in the analyzed cohort of patients (Table 3).

## 3. Discussion

Testicular GCTs have a high cure rate, even in patients with metastatic disease, due to their unique responsiveness to CDDP-based chemotherapy. However, the mechanisms underlying the pervasive growth of cancer cells in patients who are refractory to CDDP are still poorly understood [20]. Multiple pathways and factors are implicated in CDDP resistance in GCTs [21] and recent data suggest that the TME may be a critical factor in this process [4], having a regulatory role in DNA damage response [15].

In the present study, we show that PBMCs cocultivated with GCT cell lines resistant to CDDP demonstrate elevated levels of endogenous DNA damage compared to those observed in PBMCs cocultivated with their CDDP-sensitive parental GCT cell lines. Increased levels of endogenous DNA damage in PBMCs resulting from their cocultivation with CDDP-resistant cancer cells may reflect a more aggressive disease in patients who do not respond to CDDP-based chemotherapy. These data are in accordance with our previous findings showing that endogenous DNA damage levels correlate with patient prognosis independent of the IGCCCG risk group [16,17]. In both studies, the DNA damage level was measured by the comet assay and expressed as the mean percentage of DNA in the tail. Moreover, other methods are widely used to detect and quantify DNA damage in cells (including male germ cells), such as histone H2AX phosphorylation (γ-H2AX) assay [22]. γ-H2AX is currently under extensive investigation to determine whether it fulfils the requirements as a marker for oncogenic transformation. Its prognostic value has been comprehensively examined and is already indicated in certain cancer types (reviewed in [23]). We are aware of the fact that in terms of its possibility of being translated into clinical use, γ-H2AX has a substantial advantage over the comet assay, as it provides a considerably more sensitive, efficient, and reproducible measurement of the DNA damage level. In contrast to the comet assay, which possesses substantial limitations for clinical application in its present form, γ-H2AX measurement throughout immunostaining, flow cytometry or enzyme-linked immunosorbent assay is able to easily enter clinical laboratories. For this reason, studies correlating DNA damage levels in clinical samples using both the comet assay and γ-H2AX staining at the same time would be highly beneficial, as they would address a question of how the data of both assays mirror each other. Such studies are currently ongoing in our laboratory. Nevertheless, we strongly believe that the comet assay data have clinical applicability and may serve as reliable markers for many aspects of cancer biology.

Based on this knowledge as well as the results of our previous studies, we evaluated a possible crosslink between endogenous DNA damage levels and changes in the percentage of specific populations of immune cells. We found that an increased percentage of NK cells, CD16+ DCs and Tregs was independently associated with higher endogenous DNA damage levels measured in PBMCs from chemotherapy-naïve GCT patients.

CD16+ DCs represent a unique myeloid antigen-presenting cell population whose ontogeny and function are under further investigation [24]. CD16 is also known as the Fc receptor FCγRIII, which participates in signal transduction due to antibody-dependent cellular cytotoxicity [25,26]. The subpopulation of monocytes that expresses CD16 (Fcγ receptor III) is predisposed to become migratory DCs [27]. It is generally known that the majority of human malignancies are characterized by a chromosome instability phenotype that often coincides with cytosolic DNA that activates the cGAS-sensor protein stimulator of the *IFN* genes (STING) pathway, forming essential crosstalk between cancer cells and the immune microenvironment [28]. We speculate that the coincidence of elevated DNA damage levels in PBMCs and the increased percentage of CD16+ DCs might reflect the activation of this pathway.

Tregs are involved in tumor development and progression, predominantly by preventing antitumor immunity. In the TME, naïve DCs promote Treg function, thereby generating extensive bidirectional crosstalk and influencing the immune response in physiological and pathological settings [27,29]. Tregs have extensively been characterized in the PB and immune infiltrates of different cancers [30,31], where Treg infiltration correlates with poor survival [32]. Interestingly, an immunosuppressive phenotype characterized by the induction of Tregs, among others, was also determined after ultraviolet light exposure [28].

Our data also show a positive independent association between the DNA damage level and the percentage of NK cells. NK cells, as a specialized population of innate lymphoid cells, are primarily involved in controlling tumor growth and mediating a robust antimetastatic effect. Elimination of tumor cells via NK cells is induced upon binding of the NKG2D receptor to its ligand NKG2DL. This ligand is expressed on transformed cells as well as on cells with DNA damage [33,34]. The present data are in accordance with the association reported in our study: cells with DNA damage might potentially express increased levels of this ligand, which could subsequently lead to an elevation of NK cells. However, cancer cells are able to downregulate their surface ligands to obstruct antitumor recognition and escape NK cell-mediated immune surveillance [35]. In addition, the study by Alvarez et al. described NK cell exhaustion because increases in the NKG2DL levels during NK cell activation were linked to DNA damage [36].

Generally, malignant process is characterized as a heterogeneous complex disease, where accumulation of DNA damage may be a potential biomarker of genome instability during tumorigenesis and disease progression. Although there is a quite well-documented correlation between the PBMCs and tumor tissue in terms of DNA repair capacity [37]; PMBCs were suggested not to be predictive of the repair capability of the tumor, and hence not to act as surrogate cells in this context [38]. Consequently, DNA repair kinetics in PMBCs cells might rather be a consequence of manifestation of an independent cancer phenotype [25]. Logically, aim of the present in vitro study was to investigate possible interaction between the PBMCs and tumor cells in order to clarify whether PBMCs can serve as surrogate for tumor cells with respect to the prognostic value of the DNA damage level in TGCTs. Using the 2D coculture model, widely used for study of an interplay between the two cells populations [39,40,41], we have clearly shown that the DNA damage level in PBMCs cocultivated with CDDP-resistant GCT cell lines is significantly higher than in PBMCs cocultivated with their sensitive counterparts. However, we are fully aware of limitations of this in vitro setup, which is unable to simulate completely TME compared to the 3D organoid model system, and therefore further experiments are unnecessarily required to address this issue unambiguously. In addition, a small number of patients enrolled in the present study represents another limitation.

Based on the obtained results, we are not able to specify which cell subpopulation within PBMCs is the primary carrier of endogenous DNA damage. The percentage of leukocyte subpopulations (including NK cells, CD16+ DCs and Tregs) that were independently associated with the level of endogenous DNA damage is relatively underrepresented within the whole immune cell population. Therefore, we may suppose that the abovementioned subpopulations are unable to cause increased levels of DNA damage. Conversely, we may assume that changes in the percentage of selected immune cell subpopulations reflect DNA damage, as an important factor of the TME, and eventually reflect the characteristics of malignant processes. However, based on our data, we cannot determine whether this association can be characterized as causal, whether changes in the percentages of selected subpopulations of immune cells reflect differences in DNA damage levels within the TME, or whether this link represents some kind of disease manifestation. Additionally, our data do not exclude the presence of (i) common factors in the TME produced by tumor cells that induce DNA damage and simultaneously mediate changes in the percentage of selected leukocyte subpopulations or (ii) several different factors that selectively influence DNA damage and lead to alterations in the percentages of immune cell subpopulations.

In summary, we correlated the percentage of a specific population of immune cells in PB with endogenous DNA damage levels measured in PBMCs from chemotherapy-naïve GCT patients. We report that the DNA damage level is significantly associated with specific changes in the immune cell repertoire. We identified specific populations of immune cells, including specific subpopulations within NK cells, DCs and Tregs, whose percentages independently correlated with endogenous DNA damage levels. Moreover, we demonstrate that PBMCs display increased DNA damage levels after cocultivation with CDDP-resistant GCT cell lines compared to PBMCs cocultivated with their sensitive counterparts. Further research should focus on the identification of factors produced by GCTs that induce DNA damage. Finally, there is a need for a deeper understanding of specific immune cells and cell interactions in tumor surveillance that are controlled by the DNA damage response and repair.

## 4. Material and Methods

### 4.1. Cell Lines

The TCam-2 human seminoma cell line (kindly provided by Dr. Kitazawa, Ehime University Hospital, Shitsukawa, Japan) and the human ovarian yolk sac tumor cell line NOY-1 (cat. no: ENG101, FA: Kerafast) were maintained in RPMI 1640 medium (GIBCO^®^ Invitrogen, Carlsbad, CA, USA) containing 10% FBS, 10,000 IU/mL penicillin, 5 μg/mL streptomycin, 2.5 μg/mL amphotericin and 2 mM glutamine. The human embryonal carcinoma cell line NTERA-2 (ATCC^®^ CRL-1973™) was cultivated in high-glucose (4.5 g/L) DMEM (PAA Laboratories GmbH, Pasching, Austria) supplemented with 10% FBS (GIBCO^®^ Invitrogen, Carlsbad, CA, USA), 10.000 IU/mL penicillin (Biotica, Part. Lupca, Slovakia), 5 μg/mL streptomycin, 2.5 μg/mL amphotericin and 2 mM glutamine (PAA Laboratories GmbH).

CDDP-resistant derivatives of the parental GCT cell lines (TCam-2 CisR, NOY-1 CisR and NTERA-2 CisR) were obtained by long-term (6 months) cultivation of the cell lines in the presence of increasing concentrations of CDDP (Hospira UK Ltd., Warwickshire, UK), as described previously [19,20]. Briefly, cells in the exponential phase of growth were initially exposed to 0.05 µg/mL CDDP. When the cells started to expand, the concentrations were gradually increased to 0.1 µg/mL (TCam-2 CisR and NTERA-2 CisR) or to 0.3 µg/mL (NOY-1 CisR). The CDDP-resistant TCam-2 CisR and NTERA-2 CisR cell lines were continuously maintained in 0.1 µg/mL CDDP, while NOY-1 CisR cells were maintained in 0.3 µg/mL CDDP. The chemosensitivity of de novo-derived GCT CisR cell lines to CDDP was evaluated using the CellTiter-Glo Luminescent Cell Viability Assay (Promega Corporation, Madison, WI, USA). Afterwards, these cells were cultivated in medium without CDDP for an additional 4-month period. Their resistance to CDDP (expressed as the IC_50_ value) was preserved. The IC_50_ value for CDDP increased from 0.45 µg/mL in TCam-2 cells to 1.38 µg/mL in TCam-2 CisR cells, from 0.35 µg/mL in NOY-1 cells to 2.37 µg/mL in NOY-1 CisR cells and from 0.01 µg/mL in NTERA-2 cells to 0.31 µg/mL in NTERA-2 CisR cells [42,43].

### 4.2. Patients

The present prospective translational study enrolled 74 chemotherapy-naïve GCT patients treated from October 2012 to September 2019 at the National Cancer Institute and/or the St. Elisabeth Cancer Institute, Bratislava, Slovakia (Table 1). Patients with concurrent malignancies other than nonmelanoma skin cancer in the previous 5 years were excluded from this study. Clinical stage of primary disease at diagnosis was determined according to the criteria set in the Tumor Node Metastasis (TNM) staging system (2010) [44]. Data regarding age, tumor histologic subtype, clinical stage, type and number of metastatic lesions were recorded for all patients. The Institutional Review Board and Ethical Committee of the National Cancer Institute, Bratislava, Slovakia, approved the protocol (No. IZLO1; Chair: M. Mego, from 10 February 2010) used to conduct this study. All participants (including healthy subjects) provided signed informed consent before study enrollment.

### 4.3. PB Sampling

Atraumatic PB (2 mL) for cocultivation studies was collected at the antecubital fossa of healthy subjects into EDTA-treated collection tubes at baseline in the morning. For GCT patients enrolled in the study, PB was collected into lithium-heparin-treated and EDTA-treated tubes at baseline in the morning on day -1 or 0 of the first cycle of chemotherapy. One milliliter of PB treated with the anticoagulant EDTA was used as the starting material for immunophenotyping of leukocyte subpopulations, while 2 mL of PB collected into lithium-heparin-treated tubes was used for PBMC isolation and subsequently analysis by the comet assay.

### 4.4. PBMC Isolation by Density Gradient Centrifugation

PBMCs, which consisted of lymphocytes and monocytes, were isolated from EDTA- or lithium-heparin-treated blood samples using density gradient centrifugation as described previously [16]. Briefly, PB (2 mL) was diluted 1:1 with phosphate-buffered saline (PBS; 137 mM NaCl, 8 mM Na_2_HPO_4_, 2.7 mM KCl, 1.8 mM KH_2_PO_4_, pH 7.2), and the resulting mixture was carefully poured onto Histopaque-1077 (3 mL; Sigma-Aldrich, Germany). After centrifugation of the sample at 1200 rpm for 30 min at room temperature (RT), blood cells were layered, and PBMCs were separated. Subsequently, prior to cocultivation with GCT cell lines, separated PBMCs were washed twice in PBS and resuspended in RPMI 1640 culture medium (GIBCO^®^ Invitrogen, Carlsbad, CA, USA).

### 4.5. Cocultivation of GCT Cell Lines with PBMCs

GCT cell lines TCam-2, NOY-1 and NTERA-2 and their CDDP-resistant counterparts were cocultivated with PBMCs isolated from healthy subjects. PBMCs were seeded at the bottom of 12-well plates (1 × 10^5^/well) (Corning^®^ Transwell^®^ polycarbonate membrane cell culture inserts, Sigma-Aldrich, Saint-Louis, MO, USA), while GCT cell lines were seeded in 3.0 μm pore polycarbonate membrane inserts (TCam-2 1.5 × 10^5^, TCam-2 CisR 1.5 × 10^5^, NOY-1 2 × 10^5^, NOY-1 CisR 2 × 10^5^, NTERA-2 1.5 × 10^5^, NTERA-2 CisR 1.5 × 10^5^). The cells were cocultivated for 96 and 120 h (initially, 24, 48 and 72 h time points were also used) in RPMI culture medium without CDDP at 37 °C with 5% CO_2_. PBMCs cultured individually with blank inserts served as a control. Each experiment was performed in triplicate. PBMCs from each time point were collected and subsequently analyzed using the comet assay.

### 4.6. Determination of the DNA Damage Level in PBMCs

The level of DNA damage in PBMCs after cocultivation with GCT cell lines as well as in PBMCs isolated from GCT patients was determined by the comet assay. The procedure was used with minor modifications suggested by Singh et al. [45]. In brief, PBMCs embedded in 0.75% low melting point (LMP) agarose and spread on a base layer of 1% normal melting point (NMP) agarose in PBS buffer (Ca^2+^ and Mg^2+^ free) were placed in a lysis solution (2.5 M NaCl, 100 mM Na_2_EDTA, 10 mM Tris–HCl, pH 10 and 1% Triton X-100) at 4 °C for 1 h. Slides were placed in an electrophoretic box and immersed in cold lysis buffer (2.5 M NaCl, 10 mM Tris−HCl, 100 mM Na_2_EDTA, pH 10.0) containing 1% Triton-X for 60 min at 4 °C. To denature DNA, slides were placed in a horizontal gel electrophoresis tank filled with cold electrophoresis solution (1 mM Na_2_EDTA, 0.3 M NaOH, pH 13.0) for 40 min at 4 °C. Electrophoresis was performed at 0.7 V/cm, 300 mA for 30 min at 4 °C. Following electrophoresis, slides were neutralized by washing 3 times (5 min each wash) with neutralization buffer (0.4 M Tris−HCl, pH 7.5). The slides were then washed with distilled H_2_O and allowed to dry for 4 h at RT. Each slide was stained with ethidium bromide (EtBr; 20 μg/mL) for 20 min at RT. One hundred randomly selected nucleoids per slide were analyzed using Metafer-MetaCyte analysis software (Metasystems, Altlussheim, Germany), and the DNA damage was expressed as the mean % of DNA in the tail ± standard error of the mean (SEM).

### 4.7. Immunophenotyping of Leukocytes Subpopulations

PB samples (1 mL) were collected in EDTA-treated vials prior to the 1st cycle of chemotherapy and processed within 24 h. Leukocytes were labeled with fluorochrome-conjugated antibodies obtained from BD Pharmingen, and analyses were performed on a flow cytometer (Becton Dickinson Canto II Cytometer). The antibody combinations used were as follows: 1. Basic panel—CD8 FITC (clone SK1, cat. no.: 345772, BD Biosciences, San Jose, CA 95131 USA), CD56 PE (clone MY31, cat. no.: 345810, BD Biosciences, San Jose, CA 95131 USA), CD45 PerCP Cy5.5 (clone SK3, cat. no.: 332772, BD Biosciences, San Jose, CA 95131 USA), CD19 PE-Cy7 (cat. no.: IM3628, Beckman Coulter Immunotech SAS, Marseille, France), CD3 APC (clone SK7, cat. no.: 345767, BD Biosciences, San Jose, CA 95131 USA), CD16 APC-H7 (clone 3G8, cat. no.: 560195, BD Pharmingen, San Diego, CA 92121 USA), CD4 V450 (clone RPA-T4, cat. no.: 560345, BD Biosciences, San Jose, CA 95131 USA), and CD14 HV500 (clone M5E2, cat. no.: 561391, BD Biosciences, San Jose, CA 95131 USA); 2. Regulatory T cell panel—CD3 FITC (clone SK7, cat. no.: 345763, BD Biosciences, San Jose, CA 95131 USA), CD127 PE (clone hIL-7R-M21, cat. no.: 557938, BD Pharmingen, San Diego, CA 92121 USA), CD4 PerCP Cy5.5 (clone SK3, cat. no.: 566923, BD Biosciences, San Jose, CA 95131 USA), CD25 PE-Cy7 (clone 2A3, cat. no.: 335824, BD Biosciences, San Jose, CA 95131 USA), and CD45 HV450 (clone HI30, cat. no.: 560367, BD Biosciences, San Jose, CA 95131 USA); 3. Dendritic cell panel—Lin FITC (lineage cocktail 2 FITC, cat. no.: 643397, BD Biosciences, San Jose, CA 95131 USA), CD1c PE (clone F10/21A3, cat. no.: 564900, BD Pharmingen, San Diego, CA 92121 USA), HLA-DR PerCP (clone L243, cat. no.: 347402, BD Biosciences, San Jose, CA 95131 USA), CD123 PE-Cy7 (clone 7G3, cat. no.: 560826, BD Pharmingen San Diego, CA 92121 USA), CD11c APC (clone B-Ly 6, cat. no.: 560895, BD Biosciences, San Jose, CA 95131 USA), CD16 APC-H7 (clone 3G8, cat. no.: 560195, BD Pharmingen San Diego, CA 92121 USA), and CD45 HV450 (clone HI30, cat. no.: 560367, BD Biosciences, San Jose, CA 95131 USA); and 4. Myeloid-derived suppressor cell panel—CD15 FITC (cat. no.: IM1423U, Beckman Coulter Immunotech SAS, Marseille, France), CD11b PE (cat. no.: IM2581U, Beckman Coulter Coulter Immunotech SAS, Marseille, France), HLA-DR PerCP (clone L243, cat. no.: 347402, BD Biosciences, San Jose, CA 95131 USA), CD62L PE-Cy7 (clone DREG-56, cat. no.: 565535, BD Biosciences, San Jose, CA 95131 USA), CD33 APC (clone P67.6, cat. no.: 345800, BD Biosciences, San Jose, CA 95131 USA), CD14 APC-H7 (clone MΦP9, cat. no.: 641394, BD Biosciences, San Jose, CA 95131 USA), CD66b V450 (clone G10F5, cat. no.: 561649, BD Biosciences, San Jose, CA 95131 USA) and CD45 BV510 (clone 30-F11, cat. no.: 103138, Biolegend, San Diego, CA 91121 USA).

Briefly, 300,000–500,000 WBCs in 200 µL were incubated with a cocktail of monoclonal antibodies (4 tubes, each with 8 fluorochromes, defined in the section above) for 20 min at RT. Red blood cells were lysed, and cells were fixed using 2 mL of 1× BD FACS Lysing Solution (BD Bioscience, cat. no: 349202) with incubation for 10 min at RT. Subsequently, the samples were centrifuged at 200× *g* for 5 min and washed twice with PBS prior to detection. A minimum of 100,000 total leukocytes were processed on a BD FACSCanto™ II flow cytometer (Becton Dickinson, Franklin Lakes, NJ, USA), and analysis was performed using KALUZA software (Beckman Coulter). Debris was excluded according to its size and granularity using forward scatter (FSC) and side scatter (SSC), while doublets using FSC-Height and FSC-Area. The minimum number of gated cells was 100.

### 4.8. Statistical Analysis

For the statistical analysis of data obtained from the in vitro experiments, the normality assumption hypothesis was tested using the Shapiro–Wilk test. Differences between the defined groups at individual time points were assessed by Student’s *t*-test (normally distributed data).

The statistical analysis regarding patient samples included tabulation of the patient characteristics and their subsequent summarization as the median (range) values for continuous variables and frequency (percentage) for categorical variables. The Kolmogorov–Smirnov test was used to test the normality of the distribution of analyzed data. Normally distributed data were statistically evaluated by Student’s *t*-test or analysis of variance, while the nonparametric Mann–Whitney U test or the Kruskal–Wallis H test was used for nonnormally distributed data. Pearson’s or Spearman’s correlation was used according to the normality of the data.

The median follow-up period was calculated as the median observation time of all patients, including those who were still alive at the time of the last follow-up. The cutoff value of 6.34 was used for dichotomizing patients based on endogenous DNA damage level (measured as % DNA in tail) in PBMCs from individuals with chemotherapy-naïve GCT. This cutoff was calculated by receiver operator characteristic (ROC) analysis. Based on the results of ROC analysis, a value of 6.34 has a significant specificity and sensitivity to distinguish GCT patients with and those without inferior outcomes [16]. Dichotomized data were subsequently correlated with the percentage of immune cell populations by univariate analyses. Univariate analyses with the chi squared or Fisher’s exact tests were also carried out to evaluate correlations among the clinicopathological characteristics. Afterwards, a multivariate logistic regression analysis was performed with the variables identified as significantly associated with endogenous DNA damage level in univariate analysis.

All the presented *p* values are two-sided, and associations were considered significant if the *p* value was less than or equal to 0.05. Statistical analyses were performed using NCSS 11 Statistical Software (2016, NCSS, LLC., Kaysville, UT, USA, ncss.com/software/ncss, 30 October 2021).

## Figures and Tables

**Figure 1 ijms-22-08281-f001:**
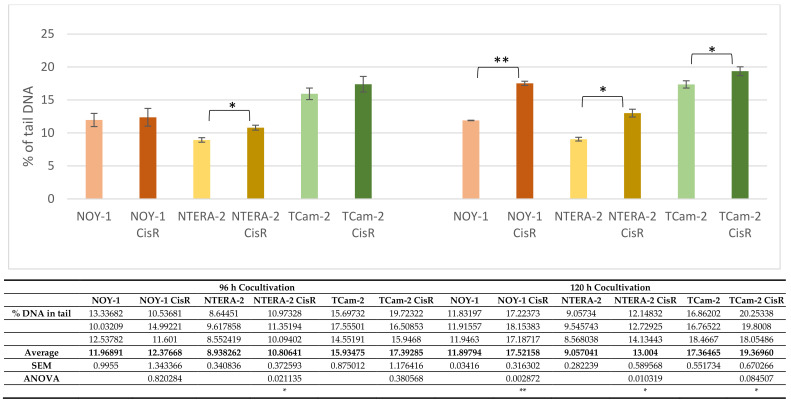
The genotoxic effects of cocultivation of GCT cell lines with PBMCs from healthy donors. The comet assay was used to determine levels of DNA damage in PBMCs after 96 and 120 h of cocultivation with the yolk sac tumor cell line NOY-1, embryonal carcinoma cell line NTERA-2, seminoma cell line TCam-2, and their CDDP-resistant (CisR) variants. Data represent the means ± SEM of three independent experiments (raw data are presented in table below Figure 1). *p* values (* *p*  <  0.05 and ** *p*  <  0.01) indicate statistically significant differences in DNA damage levels between PBMCs cocultivated with CDDP-sensitive GCT cell lines and those cocultivated with their corresponding resistant variants (ANOVA test). The table belonging to the Figure 1 summarizes raw data to the evaluation of genotoxic effects of co-cultivation of germ cell tumor cell lines and their CDDP-resistant variants with PBMCs from healthy donor.

**Figure 2 ijms-22-08281-f002:**
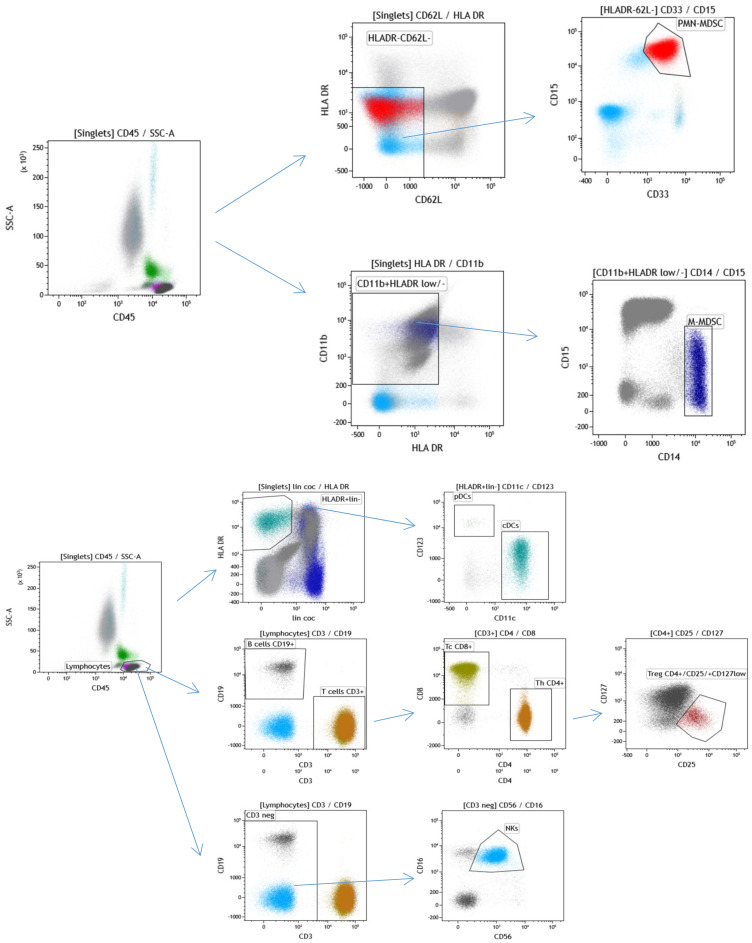
Flow cytometry gating strategy used for immunophenotyping selected leukocyte subpopulations in GCT patients. The cells were first gated by forward scatter (FSC) and side scatter (SSC) following doublet exclusion using forward scatter area (FSC-A)/forward scatter height (FSC-H) (not shown). CD15 + CD33 + CD62L-HLADR-/low (PMN-MDSCs) and CD14 + CD11b + HLADR-/low (M-MDSCs) are shown on the top. Dendritic cells were identified by CD45 + HLADR + lin- and subsequently distinguished by CD123 + CD11c-(pDC) and CD11c + (mDC) expression, as shown in the middle of the figure. Total lymphocytes were gated on a CD45/SSC and then gated on CD19+ (B cells) vs. CD3+ (T cells), CD4+ (Th cells) vs. CD8+ (Tc cells) and CD56 + CD16 + CD3-(NK cells). CD4+ cells were used for the identification of CD25+ CD127-/low (Tregs).

**Table 1 ijms-22-08281-t001:** Clinicopathological characteristics of GCT patients.

Variable	*N*	%
All patients	74	100.0
**Age (years)**		
Median (range)	35 (19–62)	
**Sex**		
Male	74	100.0
Female	0	0.0
**Histology**		
Seminoma	20	27.0
Nonseminoma	54	73.0
**Primary tumor localization**		
Testicular	71	91
Extragonadal	3	8.1
IGCCCG risk group		
Good risk	43	58.1
Intermediate risk	7	9.5
Poor risk	11	14.9
Stage IA and IB (adjuvant therapy)	13	17.6
**Sites of metastases**		
Retroperitoneum	53	71.6
Mediastinum	10	13.5
Lungs	14	18.9
Liver	6	8.1
Brain	2	2.7
Other	1	1.4
Visceral nonpulmonary metastases	8	10.8
**No. of metastatic site(s)**		
0 to 1	53	71.6
>2	21	28.4
**Staging (UICC)**		
IA	2	2.7
IB	11	14.9
IS	5	6.8
IIA	5	6.8
IIB	15	20.3
IIC	6	8.1
IIIA	8	10.8
IIIB	10	13.5
IIIC	12	16.2

IGCCCG, International Germ Cell Consensus Classification Group; UICC, Union for International Cancer Control.

**Table 2 ijms-22-08281-t002:** Association between the endogenous DNA damage level and patients’ clinicopathological characteristics using univariate statistical analysis (*N* = 74; a cutoff value of 6.34 was used for dichotomization of endogenous DNA damage level measured in the patients’ PBMCs).

		The DNA Damage Level ≤6.34 >6.34			
Variable	*N*	%	*N*	%	*p* Value ^b^
All patients	55	74.3	19	25.7	NA
**Histology**					
Seminoma	13	65.0	7	35.0	0.37
Nonseminoma	42	77.8	12	22.2	
**Tumor primary**					
Primary GCTs	54	76.1	17	23.9	0.16
Extragonadal GCTs	1	33.3	2	66.7	
**IGCCCG risk group**					
Good risk + adjuvant therapy	32	74.4	11	25.6	0.54
Intermediate + poor risk	2	28.6	5	71.4	
**No. of metastatic site(s)**					
0	14	77.8	4	22.2	0.11
1 to 2	35	79.5	9	20.5	
>3	6	50.0	6	50.0	
**Retroperitoneal lymph node metastases**					
Absent	17	81.0	4	19.0	0.56
Present	38	71.7	15	28.3	
**Mediastinal lymph node metastases**					
Absent	50	78.1	14	21.9	0.11
Present	5	50.0	5	50.0	
**Lung metastases**					
Absent	46	76.7	14	23.3	0.33
Present	9	64.3	5	35.7	
**Liver metastases**					
Absent	50	73.5	18	26.5	1.00
Present	5	83.3	1	16.7	
**Non-pulmonary visceral metastases**					
Absent	48	72.7	18	27.3	0.67
Present	7	87.5	1	12.5	
**S stage ^a^**					
0	21	80.8	5	19.2	0.61
1	19	76.0	6	24.0	
2	8	61.5	5	38.5	
3	7	70.0	3	30.0	

SEM, standard error of the mean; IGCCCG, International Germ Cell Consensus Classification Group. Values of *p* ≤ 0.05 are considered statistically significant. ^a^ Defined by the IGCCCG criteria: S0, within normal limits; S1, AFP < 1000 ng/mL and/or β-HCG < 5000 mIU/mL and/or LDH < 1.5 U/L upper normal limit; S2, AFP 1000–10,000 ng/mL and/or β-HCG 5000–50,000 mIU/mL and/or LDH 1.5–10 U/L upper normal limit; S3, AFP > 10,000 ng/mL and/or β-HCG > 50,000 IU/mL and/or LDH > 10 U/L upper normal limit. ^b^ Univariate analysis.

**Table 3 ijms-22-08281-t003:** Association between the endogenous DNA damage level and percentages of different leukocyte subpopulations in GCT patients. Univariate and multivariate logistic regression analysis was used for statistical evaluation of the obtained data.

Variable	% of Individual Leukocytes Subpopulation					
	*N*	Mean	SEM	Median	*p* Value ^b^	*p* Value ^c^
**Neutrophils percentage**						
endogenous DNA damage level ≤ 6.34	55	61.7	1.9	61.1	0.462	
endogenous DNA damage level > 6.34	19	63.9	3.2	63.4		
**Lymphocytes percentage**						
endogenous DNA damage level ≤ 6.34	54	27.5	1.7	26.6	0.232	
endogenous DNA damage level > 6.34	19	23.6	2.9	20.9		
**Monocytes percentage**						
endogenous DNA damage level ≤ 6.34	55	9.3	0.4	9.2	0.347	
endogenous DNA damage level > 6.34	19	10.2	0.7	9.6		
**Eosinophils percentage**						
endogenous DNA damage level ≤ 6.34	55	2.5	0.4	2.7	0.334	
endogenous DNA damage level > 6.34	19	3.0	0.6	2.7		
**Basophils percentage**						
endogenous DNA damage level ≤ 6.34	55	0.6	0.04	0.3	0.064	
endogenous DNA damage level > 6.34	19	0.8	0.07	0.4		
**B cells percentage**						
endogenous DNA damage level ≤ 6.34	55	12.6	0.7	11.8	0.00058	0.60005
endogenous DNA damage level > 6.34	19	8.3	1.2	8.0		
**T cells percentage**						
endogenous DNA damage level ≤ 6.34	55	72.9	1.2	73.6	0.848	
endogenous DNA damage level > 6.34	19	71.9	2.1	74.4		
**T helper cells percentage**						
endogenous DNA damage level ≤ 6.34	54	45.4	1.2	45.8	0.530	
endogenous DNA damage level > 6.34	19	43.3	2.0	43.6		
**T cytotoxic cells percentage**						
endogenous DNA damage level ≤ 6.34	55	25.6	1.0	24.5	0.669	
endogenous DNA damage level > 6.34	19	26.2	1.6	26.9		
**NKT cells percentage**						
endogenous DNA damage level ≤ 6.34	53	1.7	0.4	2.4	0.057	
endogenous DNA damage level > 6.34	19	3.1	0.6	3.3		
**CD4+ NKT cells percentage**						
endogenous DNA damage level ≤ 6.34	24	0.2	0.1	0.3	0.155	
endogenous DNA damage level > 6.34	11	0.7	0.2	0.9		
**CD8+ NKT cells percentage**						
endogenous DNA damage level ≤ 6.34	25	1.7	0.4	1.5	0.243	
endogenous DNA damage level > 6.34	11	2.7	0.5	2.4		
**NK cells percentage**						
endogenous DNA damage level ≤ 6.34	55	11.3	1.1	9.6	0.008	0.00063
endogenous DNA damage level > 6.34	19	17.3	1.9	13.6		
**Tregs percentage**						
endogenous DNA damage level ≤ 6.34	55	3.8	0.2	3.8	0.03937	0.01588
endogenous DNA damage level > 6.34	19	4.5	0.3	4.4		
**Classical monocytes percentage**						
endogenous DNA damage level ≤ 6.34	41	87.6	1.1	87.9	0.256	
endogenous DNA damage level > 6.34	17	85.7	1.7	86.5		
**Intermediate monocytes percentage**						
endogenous DNA damage level ≤ 6.34	24	4.7	0.5	4.3	0.546	
endogenous DNA damage level > 6.34	11	5.2	0.8	3.7		
**Nonclassical monocytes percentage**						
endogenous DNA damage level ≤ 6.34	40	4.5	0.5	4.8	0.714	
endogenous DNA damage level > 6.34	17	5.9	0.7	4.6		
**Polymorphonuclear leukocytes (PNMs) percentage**						
endogenous DNA damage level ≤ 6.34	36	1.0	0.5	0.3	0.0753	
endogenous DNA damage level > 6.34	15	1.1	0.8	0.2		
**Dendritic cells (cDCs) percentage**						
endogenous DNA damage level ≤ 6.34	44	0.8	0.07	0.8	0.614	
endogenous DNA damage level > 6.34	13	1.0	0.1	0.7		
**Plasmocytoid dendritic cells (pDCs) percentage**						
endogenous DNA damage level ≤ 6.34	43	0.1	0.01	0.1	0.938	
endogenous DNA damage level > 6.34	13	0.2	0.02	0.1		
**CD16+ within DCs percentage**						
endogenous DNA damage level ≤ 6.34	32	41.5	2.8	44.4	0.00574	0.00010
endogenous DNA damage level > 6.34	6	64.1	6.5	64.8		
**CD1c+ within DCs percentage**						
endogenous DNA damage level ≤ 6.34	44	23.1	1.3	22.9	0.066	
endogenous DNA damage level > 6.34	13	18.1	2.4	17.3		

SEM, standard error of the mean. Values of *p* ≤ 0.05 are considered as significant. ^b^ Univariate analysis, ^c^ Multivariate logistic regression analysis.

## Data Availability

Data is contained within the article.

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
