# Peer review of "Are Changes in the Percentage of Specific Leukocyte Subpopulations Associated with Endogenous DNA Damage Levels in Testicular Cancer Patients?"

_ijms, 2021, doi:10.3390/ijms22158281_

Round 1

Reviewer 1 Report

Kalavska et al have investigated if any specific leukocyte subpopulations that consist of the tumor microenvironment in testicular cancer are associated with endogenous DNA damage levels using a co-culture system as well as testicular cancer patient samples. There are several major issues in this manuscript.

First, the manuscript is poorly written in English, which makes it very difficult to understand. Some of the examples are in line 57-58 “ Chemoresistance of GCT is a complex multifactor-mediated process, in which the critical, but still poorly understood, role plays tumor microenvironment (TME)”. In line 71-72, “ Recent evidences demonstrate that TME of GCT…….”.  In addition, at the end of the introduction (from line 97-102), the authors just wrote that they performed in vitro analyses using the co-culture system of PBMC and then carried out more comprehensive analysis using patients’ samples. These sentences do not help the audiences to get a clear big picture regarding the contribution of this study to the testicular cancer research field or testicular cancer therapy.

Second, how the results are arranged, and communicated to the audience made it even harder to understand the major points that the authors are trying to make in this manuscript. For example, in Figure1 legend, there is no any information to assist in explaining how to get this result. In addition, there are no abbreviations nor information about NOY-1, NOY-1CisR etc. If this result is obtained using the Comet assay, the authors need to show the raw data as part of the figure. Furthermore, why the authors used 96h and 120h as the endpoint? All of these information need to be clearly communicated in the Results section. Same issue goes to all tables and figures in this manuscript.

Third, the authors did not do their job to really explain and communicate their results at all. For example, in line 107, “As obvious (Figure 2), the DNA damage level……”. This is not the language in the research publication at all. In addition, it does not make any sense at all that Figure 2 is shown at the end of the manuscript, but the authors mentioned it before where they were talking about their Figure 1 result. This is not logical at all.

Fourth, the authors used 6.34 as a cut-off value for indicating low DNA damage without giving any logical explanation. They decided to use 6.34 because this cut-off value gave them what they expected to see among the patients’ samples or this value can at least separate some immune cell subpopulations from others or based on other reasons?! The authors need to clearly include this important information because it does determine the major conclusion in this study.

Author Response

Kalavska et al have investigated if any specific leukocyte subpopulations that consist of the tumor microenvironment in testicular cancer are associated with endogenous DNA damage levels using a co-culture system as well as testicular cancer patient samples. There are several major issues in this manuscript.

First, the manuscript is poorly written in English, which makes it very difficult to understand. Some of the examples are in line 57-58 “ Chemoresistance of GCT is a complex multifactor-mediated process, in which the critical, but still poorly understood, role plays tumor microenvironment (TME)”. In line 71-72, “ Recent evidences demonstrate that TME of GCT…….”.  In addition, at the end of the introduction (from line 97-102), the authors just wrote that they performed in vitro analyses using the co-culture system of PBMC and then carried out more comprehensive analysis using patients’ samples. These sentences do not help the audiences to get a clear big picture regarding the contribution of this study to the testicular cancer research field or testicular cancer therapy.

We greatly appreciate the reviewer’s efforts and have performed the AJE language editing to improve quality of the manuscript. The selected sentences were rewritten as follows:

- line  60-61 “The chemoresistance of GCTs is a complex and multifactorial phenomenon that is closely associated with the tumor microenvironment (TME) [4].”

- line 75-78 “ The TME of GCTs is modulated by specific cytokine patterns generating proangiogenic ac-tivity in tumors. T and NK cell stimulation is reduced, which ultimately results in the immune response inhibition [11].”

To provide clear contribition of this study to the testicular cancer reasearch field and testicular cancer therapy, respectivelly, we added following sencences to the introduction:

- line 107 “Data obtained herein may contribute to elucidating the biological pathways underlying the prognostic value of endogenous DNA damage levels in peripheral lymphocytes in patients with chemotherapy-naïve testicular GCTs [16,17]. A more thorough understanding of these pathways may help to better stratify GCT patients with a high risk for relapse or poor prognosis and identify new therapeutic targets.”

Second, how the results are arranged, and communicated to the audience made it even harder to understand the major points that the authors are trying to make in this manuscript. For example, in Figure1 legend, there is no any information to assist in explaining how to get this result. In addition, there are no abbreviations nor information about NOY-1, NOY-1CisR etc. If this result is obtained using the Comet assay, the authors need to show the raw data as part of the figure. Furthermore, why the authors used 96h and 120h as the endpoint? All of these information need to be clearly communicated in the Results section. Same issue goes to all tables and figures in this manuscript.

Thank you very much for this suggestion. We agree with reviewer. To improve clarity of presented results we added suggested information to the manuscript.

The legends to the Figure 1, Figure 2 as well as to the Table 1, Table 2 and Table 3 were rewritten as follows:

Figure 1. The genotoxic effects of cocultivation of GCT cell lines with PBMCs from healthy donors. The comet assay was used to determine levels of DNA damage in PBMCs after 96 and 120 h of cocultivation with the yolk sac tumor cell line NOY-1, embryonal carcinoma cell line NTERA-2, seminoma cell line TCam-2, and their CDDP-resistant (CisR) variants. Data represent the means±SEM of three independent experiments (raw data are presented in the Table below Figure 1). P values (*p < 0.05 and **p < 0.01) indicate statistically significant differences in DNA damage levels between PBMCs cocultivated with CDDP-sensitive GCT cell lines and those cocultivated with their corresponding resistant variants (ANOVA test).

Figure 2. Flow cytometry gating strategy used for immunophenotyping selected leukocyte subpopulations in GCT patients. The cells were first gated by forward scatter (FSC) and side scatter (SSC) following doublet exclusion using forward scatter area (FSC-A)/forward scatter height (FSC-H) (not shown). CD15+CD33+CD62L- HLADR-/low (PMN-MDSCs) and CD14+CD11b+ HLADR-/low (M-MDSCs) are shown on the top. Dendritic cells were identified by CD45+HLADR+ lin- and subsequently distinguished by CD123+ CD11c- (pDC) and CD11c+ (mDC) expression, as shown in the middle of the figure. Total lymphocytes were gated on a CD45/SSC and then gated on CD19+ (B cells) vs CD3+ (T cells), CD4+ (Th cells) vs CD8+ (Tc cells) and CD56+CD16+ CD3- (NK cells). CD4+ cells were used for the identification of CD25+ CD127-/low (Tregs).     

Table 1. Clinicopathological characteristics of GCT patients.

Table 2. Association between the endogenous DNA damage level and patients’ clinicopathological characteristics using univariate statistical analysis (N = 74; a cutoff value of 6.34 was used for dichotomization of endogenous DNA damage level measured in the patients’ PBMCs).

Table 3. Association between the endogenous DNA damage level and percentages of different leukocyte subpopulations in GCT patients. Univariate and multivariate logistic regression analysis was used for statistical evaluation of the obtained data.

Furthermore, raw data to the evaluation of genotoxic effects of co-cultivation of germ cell tumor cell lines and their CDDP-resistant variants with PBMCs from healthy donor were added to the manuscript in form of the table belonging to the Figure 1.

Infomation regarding the used 96 h and 120 h endpoints were added to the Results section as follows:

“Initially, DNA damage levels were measured in PBMCs after 24, 48, 72, 92 and 120 h of coculture with GCT cell lines (data not shown). Since virtually no difference in DNA damage level was observed in PBMCs cocultured for 24, 48 and 72 h with CDDP-resistant cells compared to those cultured with CDDP--sensitive cells, only time points 96 and 120 h were used in further analyses. The time point 120 h was also established as the terminal time point, when we were able cultivate GCT cell lines without passaging.“ 

Third, the authors did not do their job to really explain and communicate their results at all. For example, in line 107, “As obvious (Figure 2), the DNA damage level……”. This is not the language in the research publication at all. In addition, it does not make any sense at all that Figure 2 is shown at the end of the manuscript, but the authors mentioned it before where they were talking about their Figure 1 result. This is not logical at all.

 Thank you very much for this point. Following your kindly suggestion, we edited sentence in the line 125 to improve its clarity:

“We determined significantly higher (p < 0.01) DNA damage levels in PBMCs cocultivated with the CDDP-resistant yolk sac tumor cell line NOY-1 CisR than in PBMCs cocultivated with CDDP-sensitive NOY-1 cells at the 120 h time point. Similarly, the DNA damage level in PBMCs cocultivated for 120 h with the CDDP-resistant embryonal carcinoma cell line NTERA-2 CisR or the seminoma cell line TCam-2 CisR was significantly higher than in PBMCs cocultivated with their sensitive counterparts (p < 0.05) (Figure 1).“

In addition, Figure 1 was incorrectly defined as Figure 2 in this sentence. We apologize for the inaccurate numbering of the abovementioned Figure which was immediately corrected.

Fourth, the authors used 6.34 as a cut-off value for indicating low DNA damage without giving any logical explanation. They decided to use 6.34 because this cut-off value gave them what they expected to see among the patients’ samples or this value can at least separate some immune cell subpopulations from others or based on other reasons?! The authors need to clearly include this important information because it does determine the major conclusion in this study.

Thank you for this point. We added suggested information to the section Material and Methods as follows:

Line 493: “The cutoff value of 6.34 was used for dichotomizing patients based on endogenous DNA damage level (measured as % DNA in tail) in PBMCs from individuals with chemotherapy-naïve GCT. This cutoff was calculated by receiver operator characteristic (ROC) analysis. Based on the results of ROC analysis, a value of 6.34 has a significant specificity and sensitivity to distinguish GCT patients with and those without inferior outcomes [16]. Dichotomized data were subsequently correlated with the percentage of immune cell populations by univariate analyses.“

Reviewer 2 Report

In the article “Are changes in percentage of specific leukocyte subpopulations associated with endogenous DNA damage levels in testicular cancer patients?” by Katarina Kalavska et collaborators aim to evaluate interplay between the immune tumor microenvironment and endogenous DNA damage level in germ cell tumors derived from patients GCT patients.

The manuscript is interesting but need to be improved to be eligible for the publication.

  1. Authors have to better describe each figure legend. In particular figure legend 1 report as <0.05 both one and two asterisks.
  2. Table 1 should report the demographic characteristics of patients (age, sex etc..)
  3. Table 2 and 3 have formatting errors and are bad of quality. Furthermore authors should decide to report or SD or SEM for data
  4. DNA damage should be represented also with H2aX staining (i.e. see the article J Cell Sci. 2018 Mar 20;131(6):jcs214411. doi: 10.1242/jcs.214411). If not possible please write as limit of the article.

Author Response

In the article “Are changes in percentage of specific leukocyte subpopulations associated with endogenous DNA damage levels in testicular cancer patients?” by Katarina Kalavska et collaborators aim to evaluate interplay between the immune tumor microenvironment and endogenous DNA damage level in germ cell tumors derived from patients GCT patients.

The manuscript is interesting but need to be improved to be eligible for the publication.

Authors have to better describe each figure legend. In particular figure legend 1 report as <0.05 both one and two asterisks.

Thank you very much for this point. We added suggested points to the manuscript.

The legends to the Figure 1  and Figure 2 were rewritten as follows:

Figure 1. The genotoxic effects of cocultivation of GCT cell lines with PBMCs from healthy donors. The comet assay was used to determine levels of DNA damage in PBMCs after 96 and 120 h of cocultivation with the yolk sac tumor cell line NOY-1, embryonal carcinoma cell line NTERA-2, seminoma cell line TCam-2, and their CDDP-resistant (CisR) variants. Data represent the means±SEM of three independent experiments (raw data are presented in the Table below Figure 1). P values (*p < 0.05 and **p < 0.01) indicate statistically significant differences in DNA damage levels between PBMCs cocultivated with CDDP-sensitive GCT cell lines and those cocultivated with their corresponding resistant variants (ANOVA test).

Figure 2. Flow cytometry gating strategy used for immunophenotyping selected leukocyte subpopulations in GCT patients. The cells were first gated by forward scatter (FSC) and side scatter (SSC) following doublet exclusion using forward scatter area (FSC-A)/forward scatter height (FSC-H) (not shown). CD15+CD33+CD62L- HLADR-/low (PMN-MDSCs) and CD14+CD11b+ HLADR-/low (M-MDSCs) are shown on the top. Dendritic cells were identified by CD45+HLADR+ lin- and subsequently distinguished by CD123+ CD11c- (pDC) and CD11c+ (mDC) expression, as shown in the middle of the figure. Total lymphocytes were gated on a CD45/SSC and then gated on CD19+ (B cells) vs CD3+ (T cells), CD4+ (Th cells) vs CD8+ (Tc cells) and CD56+CD16+ CD3- (NK cells). CD4+ cells were used for the identification of CD25+ CD127-/low (Tregs).     

Table 1 should report the demographic characteristics of patients (age, sex etc..)

Thank you very much for this point. We added demografic characteristics of the patients enrolled into the study, including data regarding age (median, range) and sex to the Table 1.  

Table 2 and 3 have formatting errors and are bad of quality. Furthermore authors should decide to report or SD or SEM for data

We much appreciate the reviewer’s careful review. Incorrect format of the Table 2 and Table 3 was caused by automatic formatting tools defined by journal. We apologize for the inaccurate formatting of the abovementioned Tables which was immediately corrected.

In addtion, only SEM is actually reported for data presented in the whole manuscript.

DNA damage should be represented also with H2aX staining (i.e. see the article J Cell Sci. 2018 Mar 20;131(6):jcs214411. doi: 10.1242/jcs.214411). If not possible please write as limit of the article.

We absolutely agree that DNA damage level is also very often measured using γH2AX staining. We have included additional paragraph into Discussion section addressing this issue and disclosing limitations our study may have.

These data are in accordance with our previous findings showing that endogenous DNA damage levels correlate with patient prognosis independent of the IGCCCG risk group [16,17]. In both studies, the DNA damage level was measured by the comet assay and expressed as the mean percentage of DNA in the tail. Moreover, other methods are widely used to detect and quantify DNA damage in cells (including male germ cells), such as histone H2AX phosphorylation (γ-H2AX) assays [22]. γ-H2AX is currently under extensive investigation to determine whether it fulfils the requirements as a marker for oncogenic transformation. Its prognostic value has been comprehensively examined and is already indicated in certain cancer types [reviewed in 23]. We are aware of the fact that in terms of its possibility of being translated into clinical use, γ-H2AX has a substantial advantage over the comet assay, as it provides a considerably more sensitive, efficient, and reproducible measurement of the DNA damage level. In contrast to the comet assay, which possesses substantial limitations for clinical application in its present form, γ-H2AX measurement throughout immunostaining, flow cytometry or enzyme-linked immunosorbent assay is able to easily enter clinical laboratories. For this reason, studies correlating DNA damage levels in clinical samples using both the comet assay and γ-H2AX staining at the same time would be highly beneficial, as they would address a question of how the data of both assays mirror each other. Such studies are currently ongoing in our laboratory. Nevertheless, we strongly believe that the comet assay data have clinical applicability and may serve as reliable markers for many aspects of cancer biology.

Round 2

Reviewer 1 Report

In this revised version of the manuscript, the authors have significantly improved their writing and added more necessary information to the readers in the content. However, there are still a few remained concerns especially in the design of the experiments in the vitro study section. In addition, the way to show the results section without a subtitle to summarize the findings is inappropriate. For example, in lines 111-113, lines 112-113 must include the summary of the in vitro study to help readers reach the conclusion clearly and quickly as possible. Same for the clinical study.

Second, in the abstract, lines 29-30, the authors wrote "Chemoresistance of  germ cell tumors is a complex multifactorial process". This statement is sort of confusing because chemoresistance is a property for germ cell tumors instead of a process. However, germ cell tumors when treated with a chemical over time do undergo a complicated process to obtain their chemoresistance property. This needs to be clarified. 

Third, the authors did talk about the tumor environment (TME) since their abstract. However, in the in vitro study when the authors co-cultured cisplatin resistant or sensitive cell lines with the PBMCs from health individuals, they only used 2D co-culture system according to the description in the Materials and Methods. This is inconsistent with their TME claim or what they wanted to pursue. Because 2D culture without using any ECM apparently not only affects the cell environment, but also fails to mimic physiological or pathological environment especially in this manuscript that the authors are trying to link the in vitro data to the in vivo (patients') results. In addition, in the manuscript, the authors mentioned that the mechanisms leading to chemoresistance observed in germ cell tumors remain unclear, whereas their in vitro study setting is not really contributing to identifying these mechanisms at all given that 2D co-culture system has no capability to address the cell environment nor the tumor microenvironment. 

Finally, it is abnormal that the authors placed a figure (Fig 2) in the Materials and Methods section. If the authors think this figure is critical, it should be move to the results section. Otherwise, it should be included in the supplemental figure or the authors can cite a reference to their previous publication. 

Author Response

In this revised version of the manuscript, the authors have significantly improved their writing and added more necessary information to the readers in the content. However, there are still a few remained concerns especially in the design of the experiments in the vitro study section. In addition, the way to show the results section without a subtitle to summarize the findings is inappropriate. For example, in lines 111-113, lines 112-113 must include the summary of the in vitro study to help readers reach the conclusion clearly and quickly as possible. Same for the clinical study.

We much appreciate the reviewer’s careful review. Following your kindly suggestion, to the result section were added subtitles dividing this section to subsections evaluating the in vitro study as well as the clinical study separately. We also included the summary for the abovementioned sections as follows:

" The genotoxic effects resulting from coculturing GCT cell lines with PBMCs from healthy donors were evaluated by the comet assay and expressed as a % of tail DNA. The DNA damage level in PBMCs cocultivated with cisplatin (CDDP)-resistant GCT cell lines was significantly higher when compared with PBMCs cocultivated with their sensitive coun-terparts after 120 h of cocultivation. "

" Our analysis revealed that endogenous DNA damage levels above the cutoff value are independently associated with increased percentages of NK cells, CD16-positive DCs and Tregs. "

Second, in the abstract, lines 29-30, the authors wrote "Chemoresistance of  germ cell tumors is a complex multifactorial process". This statement is sort of confusing because chemoresistance is a property for germ cell tumors instead of a process. However, germ cell tumors when treated with a chemical over time do undergo a complicated process to obtain their chemoresistance property. This needs to be clarified.

Thank you very much for this suggestion. We agree with reviewer. To improve clarity of the statement (lines 29-30), we rewrote it as follows:

"Chemoresistance of germ cell tumors (GCTs) represents intensive studied property of GCTs that is result of a complicated multifactorial process."

Third, the authors did talk about the tumor environment (TME) since their abstract. However, in the in vitro study when the authors co-cultured cisplatin resistant or sensitive cell lines with the PBMCs from health individuals, they only used 2D co-culture system according to the description in the Materials and Methods. This is inconsistent with their TME claim or what they wanted to pursue. Because 2D culture without using any ECM apparently not only affects the cell environment, but also fails to mimic physiological or pathological environment especially in this manuscript that the authors are trying to link the in vitro data to the in vivo (patients') results. In addition, in the manuscript, the authors mentioned that the mechanisms leading to chemoresistance observed in germ cell tumors remain unclear, whereas their in vitro study setting is not really contributing to identifying these mechanisms at all given that 2D co-culture system has no capability to address the cell environment nor the tumor microenvironment.

Thank you very much for this point. The primary aim of in vitro study was to investigate possible interaction between PBMCs and tumor cells. The 2D co-culture system is used for study of interplay between two cells populations (Chang, David H et al. “The effect of lung cancer on cytokine expression in peripheral blood mononuclear cells.” PloS one vol. 8,6 e64456. 6 Jun. 2013, doi:10.1371/journal.pone.0064456; Babini, Gabriele et al. “A Co-culture Method to Investigate the Crosstalk Between X-ray Irradiated Caco-2 Cells and PBMC.” Journal of visualized experiments : JoVE ,131 56908. 30 Jan. 2018, doi:10.3791/56908; Prieto-García, Elena et al. “Tumor-Stromal Interactions in a Co-Culture Model of Human Pancreatic Adenocarcinoma Cells and Fibroblasts and Their Connection with Tumor Spread.” Biomedicines vol. 9,4 364. 31 Mar. 2021, doi:10.3390/biomedicines9040364). However, we agree with the reviewer, that 3D organoid system (using ECM) represent more appropriate model system for study of TME.

Therefore, we included additional paragraph into Discussion section addressing this issue as limitation of our study.

" Generally, the malignant process is characterized as a heterogeneous complex disease, where the accumulation of DNA damage may be a potential biomarker of genome instability during tumorigenesis and disease progression. Although there is a well-documented correlation between the lymphocytes and tumor tissue as to DNA repair capacity [37], nevertheless lymphocytes were suggested not to solely represent surrogate cells in this context [38]. Consequently, DNA repair dynamics in PB cells might rather be a consequence of manifestation of an independent cancer phenotype [25]. Logically, aim of the present in vitro study was to investigate possible interaction between the PBMCs and tumor cells, and thus to clarify whether PBMCs can serve as a surrogate for tumor cells with respect to prognostic value of the DNA damage level in TGCTs. Using the 2D coculture model, widely used for study of an interplay between the two cells populations [39, 40, 41], we have clearly shown that the DNA damage level in PBMCs cocultivated with CDDP-resistant GCT cell lines is significantly higher than in PBMCs cocultivated with their sensitive counterparts. Despite the data unambiguous obtained, we are fully aware of limitations of this in vitro setup, which is unable to simulate completely TME compared to the 3D organoid model system, and therefore further experiments are unnecessarily required to address this issue solidly. In addition, a small number of patients enrolled in the study represents other limitation of the present study."

Finally, it is abnormal that the authors placed a figure (Fig 2) in the Materials and Methods section. If the authors think this figure is critical, it should be move to the results section. Otherwise, it should be included in the supplemental figure or the authors can cite a reference to their previous publication.

Thank you very much for this suggestion. We agree with reviewer and moved the Figure 2 to the results section.

Reviewer 2 Report

Authors reply to reviewers questions. Manuscript is eligible for publication

Author Response

We much appreciate the reviewer’s careful review

Round 3

Reviewer 1 Report

In this revised reversion, the authors have significantly improved their manuscript according to the suggestions. However, there are couples issues remaining. First, the authors still missed to address the previously mentioned issue regarding the conclusion for their Results section. The statement for their in vitro study (line 111) stayed exactly same as their previous version. There is no conclusion from their in vitro study as the title written at line 111. With this current non-conclusive subtitle for their in vitro study makes it confusing especially the in vitro study setting did not reflect the pathological condition to address the TME effect even though the authors did discuss this limitation in their Discussion section. However, this part still significantly disconnected from the clinical results/data. 

In the discussion part, line 320 states "DNA repair dynamics in PB cells might rather be a consequence....". Is this PB cells correct or it should be PBMCs? If it is PB cells, please clarify by its full name or give a brief introduction because this is the first time to see PB cells in this revised manuscript and it disconnects from the previous sentences which is very confusing. In addition, this newly written paragraph (lines 315-332) needs to be rechecked with the help from a native English speaker because several sentences do not make sense at all.

Author Response

In this revised reversion, the authors have significantly improved their manuscript according to the suggestions. However, there are couples issues remaining. First, the authors still missed to address the previously mentioned issue regarding the conclusion for their Results section. The statement for their in vitro study (line 111) stayed exactly same as their previous version. There is no conclusion from their in vitro study as the title written at line 111. With this current non-conclusive subtitle for their in vitro study makes it confusing especially the in vitro study setting did not reflect the pathological condition to address the TME effect even though the authors did discuss this limitation in their Discussion section. However, this part still significantly disconnected from the clinical results/data. 

Thank you very much for this point. In our in vitro study was primary focused to reveal the potential interaction between PBMCs and GCTs cells, mimicking the interplay between tumor cells and immune cells in GCTs patients. Accoring to the published works (Chang, David H et al. “The effect of lung cancer on cytokine expression in peripheral blood mononuclear cells.” PloS one vol. 8,6 e64456. 6 Jun. 2013, doi:10.1371/journal.pone.0064456; Babini, Gabriele et al. “A Co-culture Method to Investigate the Crosstalk Between X-ray Irradiated Caco-2 Cells and PBMC.” Journal of visualized experiments : JoVE ,131 56908. 30 Jan. 2018, doi:10.3791/56908; Prieto-García, Elena et al. “Tumor-Stromal Interactions in a Co-Culture Model of Human Pancreatic Adenocarcinoma Cells and Fibroblasts and Their Connection with Tumor Spread.” Biomedicines vol. 9,4 364. 31 Mar. 2021, doi:10.3390/biomedicines9040364) we have used the 2D co-culture system as the model system partially able to simulate this interplay. However, we agree with the reviewer, that 3D organoid system (using ECM) represent more appropriate model system for study of TME.

According your kindly suggestion, the subsection “2D Cocultivation of GCT cell lines with PBMCs isolated from healthy donor“ was changed as follows:

2D coculture model system was used to investigate possible interaction between PBMCs and tumor cells and their CDDP-resistant variants, respectively and so partial-ly mimic the interplay between tumor cells and PBMCs in GCTs patients. The geno-toxic effects resulting from coculturing GCT cell lines with PBMCs were evaluated by the comet assay and expressed as a % of tail DNA. Applying the 2D coculture model we have achieved interesting results showing that the DNA damage level in PBMCs cocultivated with cisplatin (CDDP)-resistant GCT cell lines is significantly higher than in PBMCs cocultivated with their sensitive counterparts after 120 h of cocultivation. These data suggest that CDDP-resistant tumor cells interact with PBMCs in a different way, when compared to cisplatin-sensitive tumor cells. However, we also accept the incomplete approximation of the in vitro setups, that are not fully able to simulate TME when compared to the 3D organoid model system (see limitations of the study).

In the discussion part, line 320 states "DNA repair dynamics in PB cells might rather be a consequence....". Is this PB cells correct or it should be PBMCs? If it is PB cells, please clarify by its full name or give a brief introduction because this is the first time to see PB cells in this revised manuscript and it disconnects from the previous sentences which is very confusing. In addition, this newly written paragraph (lines 315-332) needs to be rechecked with the help from a native English speaker because several sentences do not make sense at all.

Thank you very much for this point. According your kindly suggestion, the discussion part (lines 317-335) was rewritten as follows:

Generally, malignant process is characterized as a heterogeneous complex disease, where accumulation of DNA damage may be a potential biomarker of genome instability during tumorigenesis and disease progression. Although there is quite well-documented correlation between the PBMCs and tumor tissue in terms of DNA repair capacity [37], PMBCs were suggested not to be predictive of the repair capability of the tumor, and hence not to act as surrogate cells in this context [38]. Consequently, DNA repair kinetics in PMBCs cells might rather be a consequence of manifestation of an independent cancer phenotype [25]. Logically, aim of the present in vitro study was to investigate possible interaction between the PBMCs and tumor cells in order to clarify whether PBMCs can serve as surrogate for tumor cells with respect to prognostic value of the DNA damage level in TGCTs. Using the 2D coculture model, widely used for study of an interplay between the two cells populations [39, 40, 41], we have clearly shown that the DNA damage level in PBMCs cocultivated with CDDP-resistant GCT cell lines is significantly higher than in PBMCs cocultivated with their sensitive counterparts. However, we are fully aware of limitations of this in vitro setup, which is unable to simulate completely TME compared to the 3D organoid model system, and therefore further experiments are unnecessarily required to address this issue unambiguously. In addition, a small number of patients enrolled in the present study represents another limitation